# FROM PREDICTION TO PERFECTION: INTRODUCING REFINEMENT TO AUTOREGRESSIVE IMAGE GENERATION

**Cheng Cheng**[1,4,*]   **Lin Song**[4,*]   **Di An**[2]   **Yicheng Xiao**[3]
**Xuchong Zhang**[1,†]   **Hongbin Sun**[1]   **Ying Shan**[4]
[1] State Key Laboratory of Human-Machine Hybrid Augmented Intelligence,
    and Institute of Artificial Intelligence and Robotics,
    Xi'an Jiaotong University
[2] Johns Hopkins University    [3] Tsinghua University
[4] ARC Lab, Tencent PCG
`cheng2016@stu.xjtu.edu.cn`

## ABSTRACT

Autoregressive (AR) models have emerged as a powerful framework for image generation, yet they remain bound by a fundamental limitation: once a prediction is made, it cannot be revised. Each step marches forward in a strict left-to-right sequence, causing small errors to accumulate and compromise the final image. In this work, we reimagine this process with **TensorAR**, a decoder-only AR model that shifts from predicting discrete tokens to predicting overlapping *tensors*, which are essentially several adjacent discrete image tokens. This simple change transforms image synthesis into a process of *next-tensor prediction*, enabling the model to refine earlier outputs while preserving the causal structure that defines autoregression. To guard against information leakage during training, we introduce a discrete tensor noising mechanism inspired by discrete diffusion theory, which injects categorical noise into input tensors. TensorAR is designed to be plug-and-play: unlike masked AR methods, it requires no architectural modifications, and unlike autoregressive diffusion, it preserves the familiar AR training paradigm. We evaluate TensorAR across both class-to-image and text-to-image tasks, showing consistent gains in generation quality and instruction-following ability, while achieving a superior balance between quality and latency. In doing so, TensorAR offers a new path forward for autoregressive generation—one where predictions are not just produced, but continually refined.

## 1 INTRODUCTION

Building on the exceptional success of autoregressive (AR) models in natural language processing, attributable to their scalability, flexibility, and capacity to capture complex sequential dependencies, researchers have extended AR approaches to conditional image generation and to unified understanding and generation frameworks (Pang et al., 2024; Yu et al., 2024; Sun et al., 2024; Luo et al., 2024; Yu et al., 2023; Tian et al., 2024; Li et al., 2024a; Esser et al., 2021; Lee et al., 2022; Xiao et al., 2025b;a). At their core, AR models rely on a simple yet effective self-supervised objective: predicting the next token in a sequence. Compared with other generation paradigms (e.g., flow-matching models), AR models enable structured, step-by-step synthesis and offer advantages in controllability and multimodal integration (Wu et al., 2024; Team, 2024).

For image generation tasks, standard AR models (Pang et al., 2024; Yu et al., 2024; Sun et al., 2024) typically serialize images by treating each image patch as a discrete token and modeling dependencies in a predefined order (e.g., a raster scan). This paradigm forces prediction in a counter-intuitive sequence order that disrupts spatial continuity; early tokens are often blurry, which can degrade

---

[*]Equal contribution.
[†]Corresponding Author: zhangxc0329@xjtu.edu.cn.

overall quality. To improve AR generation quality, a variety of approaches have been proposed, including combining AR with continuous diffusion (Gu et al., 2024; Deng et al., 2024), modeling per-token probability distributions (Li et al., 2024a; Fan et al., 2024), and exploring alternative generation paradigms (Tian et al., 2024; Ren et al., 2025). For example, MAR (Li et al., 2024a) models per-token probability distributions via a diffusion procedure, enabling AR models to operate in continuous space and eliminating the need for discrete tokenizers. DART (Gu et al., 2024) unifies autoregression and diffusion within a non-Markovian framework, iteratively denoising image patches across spatial and spectral dimensions using an AR model with a standard language-model architecture. VAR (Tian et al., 2024) adopts a next-scale prediction framework that emulates human sketching through coarse-to-fine, 2D-parallel generation. Despite strong results, these methods typically require additional VQ-VAE training or a modification in training objective (from classification to regression), which increases computational and memory costs and may hinder multimodal integration. Parallel to these existing works, motivated by the coarse-to-fine principle that underpins diffusion and flow-matching models, we ask: *Can existing standard AR models be enabled to refine their own predictions without modifying their architecture or training recipe?*

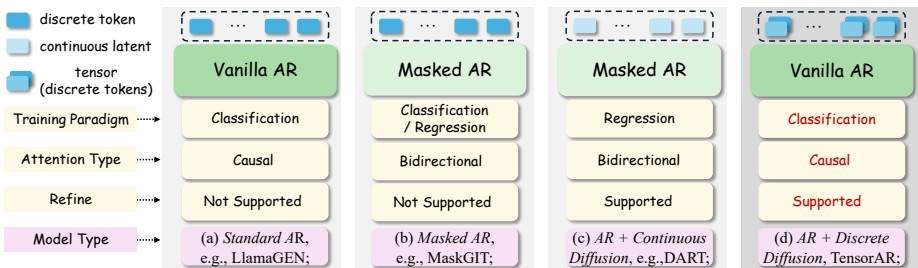

Figure 1: Comparison with different AR-based methods. (a) Vanilla AR models that directly perform next-token-prediction; (b) Masked AR models that predict masked tokens given clean tokens; (c) Integration with diffusion models that utilize the continuous output latent of AR models as the condition to an additional diffusion generation head; (d) The proposed TensorAR that does not modify the base architecture and classification-based training paradigm.

In this paper, we introduce *TensorAR*, a coarse-to-fine autoregressive image generation framework that reframes the conventional next-token prediction paradigm as "*next-tensor-prediction*". The core idea behind TensorAR is simple. Unlike standard AR models that generate one token at a time, TensorAR predicts a tensor, i.e., a group of consecutive tokens, at each step, which is the origin of the name, i.e., TensorAR. Because adjacent tensors overlap, later predictions can revise earlier ones, enabling iterative refinement of image content similar to diffusion models. For clarity, we provide a visual comparison in Figure 1. Unlike masked AR models, TensorAR does not require architectural modifications, and unlike autoregressive diffusion models, it does not alter the training paradigm.

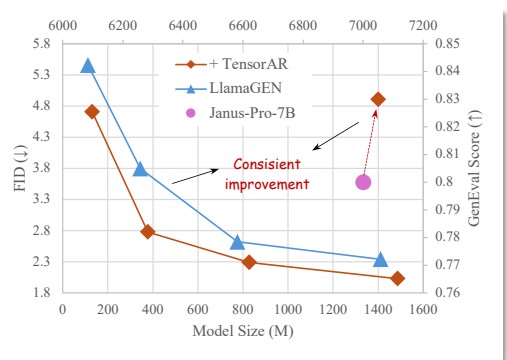

Figure 2: Model size-FID curves on TensorAR across different tasks. TensorAR achieves consistent improvements on both class-to-image and text-to-image generation tasks. Best view in color.

However, training TensorAR is nontrivial. A naive strategy would mimic standard AR training by feeding a sequence of ground-truth tensors and supervising the prediction of next-step tensors. Nevertheless, because tensors are generated in a sliding-window fashion, some tokens in the predicted tensor already appear in the input tensors, causing information leakage, where the model can minimize loss by copying overlapping tokens rather than learning meaningful causal dependencies. To address this, we introduce a discrete tensor noising mechanism based on discrete diffusion theory, which injects categorical noise into input

tensors during training. By modulating noise levels token-wise within each tensor, we stimulate an internal progressive denoising process in TensorAR. In addition, we incorporate two lightweight modules, i.e., an input encoder and an output decoder, to interface with tensor-based inputs and outputs. Both modules use the residual design to better leverage pretrained models and promote faster, more stable convergence. Together, these components make TensorAR a plug-and-play extension that integrates with existing AR models with minimal changes to the base architecture, improving practical flexibility relative to training from scratch. We evaluate TensorAR on representative AR models for class-conditional (e.g., LlamaGen (Sun et al., 2024)) and text-conditional (e.g., Janus-Pro-7B) image generation across multiple model sizes. We conduct extensive experiments across a range of base models and model sizes and comprehensive ablation studies, consistent performance gains on both tasks (Figure 2) confirm the effectiveness of the refinement mechanism and show a better trade-off between quality and latency.

## 2 TENSORAR

In this section, we first revisit the details about autoregressive modeling and discrete diffusion in 2.1 and then provide detailed explanations of our proposed method in 2.2.

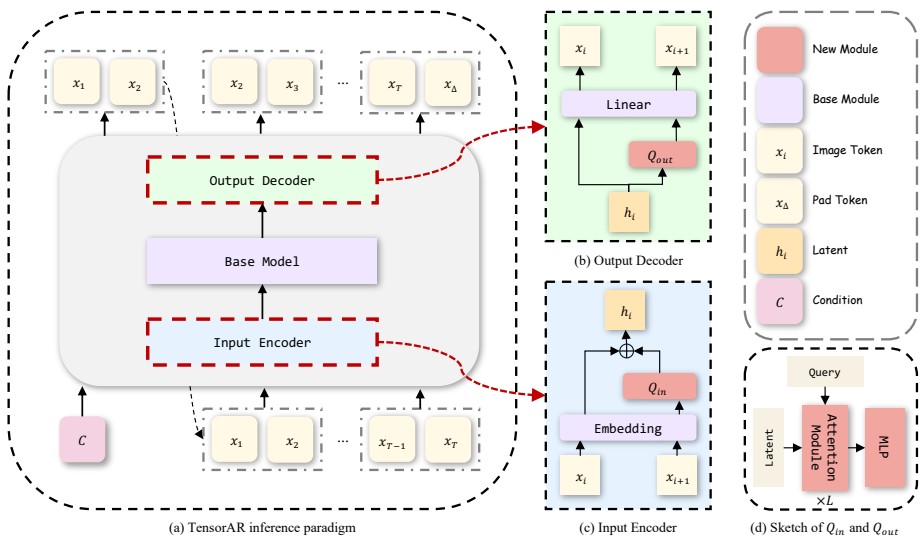

Figure 3: (a) Overview of our proposed TensorAR framework during inference time with the window size $k = 2$ and the sequence length $T$; (b) Output decoder that wraps the original linear output layer with residual design; (c) Input encoder that wraps the original embedding layer with residual design; (d) Sketch of $Q_{in}$ and $Q_{out}$, which can be implemented by query transformers. The newly introduced modules are colored in orange and the base modules are in purple.

### 2.1 PRELIMINARIES

In the following paragraph, we use $\mathbf{x}$ to denote a sequence of discrete tokens; $x$ denotes one discrete token; $\boldsymbol{x}$ denotes the one-hot version of $x$; $x^*$ denotes the noisy token of $x$.

#### 2.1.1 AUTOREGRESSIVE IMAGE GENERATION

Given a sequence of discrete tokens $\mathbf{x} = [x_1, x_2, ..., x_T]$ of length $T$ and its condition $c$, where $x_i \in \{0, 1, ..., C-1\}$ is an integer from a vocabulary of size $C$, an autoregressive model $\zeta_\theta$ are trained to model the probability distribution of each variable $x_t$ based on on its precedents $[x_1, x_2, ..., x_{t-1}]$:
$\zeta_\theta(\mathbf{x}; c) = \prod_{t=1}^{T} \zeta_\theta(x_t | x_1, ..., x_{t-1}; c)$, where $c$ may be either class labels or textual prompts, and $\zeta_\theta$ is the token distribution predictor with a model parameterized by $\theta$.

To apply autoregressive modeling to 2D images, images are first tokenized into several discrete tokens via a pre-defined order, where each discrete token corresponds to an image patch. Given $p_{\text{data}}$ as the distribution of discrete image data, the training objective of autoregressive models is to minimize the negative log-likelihood loss, which is formulated as:

$$\mathcal{L}(\theta) = \mathbb{E}_{x_{1:T} \sim p_{\text{data}}} \Big[ - \sum_{t=1}^{T} \log \zeta_\theta(x_t \mid x_{<t}, c) \Big]. \tag{1}$$

### 2.1.2 DISCRETE DIFFUSION

Discrete diffusion models (Sohl-Dickstein et al., 2015; Hoogeboom et al., 2021; Austin et al., 2021) are a class of latent variable models characterized by a forward noising process and a learned reverse denoising process. The forward process $q(\boldsymbol{x}_{1:T}|\boldsymbol{x}_0) = \prod_{t=1}^{T} q(\boldsymbol{x}_t|\boldsymbol{x}_{t-1})$ corrupts the original data $\boldsymbol{x}_0$ into a sequence of increasingly noisy latent variables $\boldsymbol{x}_{1:T}$. The backward process learns to gradually denoise the latent variables of the data distribution as $p_\theta(\boldsymbol{x}_{0:T}) = p(\boldsymbol{x}_T) \prod_{t=1}^{T} p_\theta(\boldsymbol{x}_{t-1}|\boldsymbol{x}_t)$.

According to existing studies (Zheng et al., 2023), by defining both the forward and backward distribution as categorical distribution, i.e., $q(\boldsymbol{x}_t|\boldsymbol{x}_{t-1}) = \text{Cat}(\boldsymbol{x}_t; p = \boldsymbol{Q}_t \boldsymbol{x}_{t-1})$, where $\text{Cat}(\boldsymbol{x}|p)$ is a categorical distribution over the one-hot vector $\boldsymbol{x}$ with probabilities given by the vector $p$ and $\boldsymbol{Q}_t$ is the time-dependent transition matrix, the forward process posterior $q(\boldsymbol{x}_{t-1}|\boldsymbol{x}_t, \boldsymbol{x}_0)$ and the optimization objectives can be calculated analytically, which is simply as a weighted cross-entropy loss.

$$\mathcal{L}(\theta) = \mathbb{E}_{\boldsymbol{x}_0 \sim p_{\text{data}}, \, t \sim \gamma(t), \, \boldsymbol{x}_t \sim q(\boldsymbol{x}_t|\boldsymbol{x}_0, t)} \Big[ - w_t \log p_\theta(\boldsymbol{x}_0 \mid \boldsymbol{x}_t, t) \Big], \tag{2}$$

where $p_{\text{data}}$ is the true data distribution, $t$ is the noise timestep calculated by the scheduling function $\gamma(\cdot)$, $w_t$ is the weighting coefficient.

## 2.2 TENSORAR

### 2.2.1 OVERALL FRAMEWORK

TensorAR serves as a plug-and-play module compatible with existing transformer-based autoregressive models. Unlike standard AR models that operate on sequences of tokens, TensorAR operates on sequences of *tensors*. To this end, TensorAR rearranges the sequence of tokens $\mathbf{x} = [x_1, x_2, ..., x_T]$ into the sequence of overlapping tensors $\mathbf{x}_k = [\mathbf{x}_{1,k}, \mathbf{x}_{2,k}, ..., \mathbf{x}_{T,k}]$, where $\mathbf{x}_{i,k} = [x_i, x_{i+1}, ..., x_{i+k-1}]$ is a single tensor with $k$ being its the window size. It is worth noting that an additional padding token $x_\Delta$ is added in the last few tensors of $\mathbf{x}_k$, as shown in Figure 3. During training, we ignore the loss on these padding tokens, while during inference, these padding tokens do not contribute to the final results. By reformulating the original Markov process over a token sequence into a Markov process over a tensor sequence, TensorAR adopts the *next-tensor generation* paradigm, which can be expressed as:

$$p_\theta(\mathbf{x}_k; c) = \prod_{t=1}^{T} p_\theta(\mathbf{x}_{t,k}|\mathbf{x}_{1,k}, ..., \mathbf{x}_{t-1,k}; c); \quad \mathbf{x}_{i,k} = [x_i, x_{i+1}, ..., x_{i+k-1}]. \tag{3}$$

### 2.2.2 REFINEMENT MECHANISM

The major advantage of TensorAR is its ability to refine previously generated tokens, a capability that standard autoregressive models lack. Consider a predicted tensor $\mathbf{x}_{i,k}$, within this tensor, the first token $x_i$ is the most refined, having undergone $k$ refinement steps, whereas the last token $x_{i+k-1}$ has been produced only once. Consequently, the corresponding image patch is expected to exhibit finer-grained details as the number of refinement steps increases. Intuitively, TensorAR decodes image patches iteratively in a coarse-to-fine manner, whereas standard AR methods generate each patch once in a single pass. This paradigm enables TensorAR to more effectively exploit future context to refine earlier content, resulting in higher generation quality.

As shown in Figure 3 (d), to accommodate tensor-based inputs and outputs, TensorAR introduces an input encoder $M_{enc}$ and an output decoder $M_{dec}$ that wrap the original embedding and linear output layers, respectively. The input encoder compresses several token embeddings into one single hidden

state, while the output decoder reconstructs several consecutive tokens from one single hidden state. Specifically, compression and decompression are performed by two additional modules, $Q_{in}$ and $Q_{out}$, respectively. These modules share a similar architecture and can be implemented with query transformers, which contain an attention module with several cross-attention layers and one output MLP module. Moreover, to better leverage pretrained models and to facilitate stable convergence during early training, we incorporate a residual mechanism into both $M_{enc}$ and $M_{dec}$.

### 2.2.3 NOISE MECHANISM

As shown in Figure 3 (a), considering the overlapping tokens during training, directly applying autoregressive models to tensor sequences encounters the information leakage problem, as some tokens in the predicted tensor already appear in the input tensor. This causes the model to collapse into simply replicating the overlapping tokens, rather than learning meaningful dependencies.

To address this issue, inspired by discrete diffusion theory, we propose the discrete tensor noising scheme, which adds noise to the input tensors during training. Let us begin with a simple case with a tensor $(x_i, x^*_{i+1}, ..., x^*_{i+k-1})$ where the superscript $*$ represents noisy tokens. During training time, the ideal output will be a tensor of clean tokens $(x_{i+1}, ..., x_{i+k})$. Therefore, for the overlapping tokens, TensorAR serves as the *denoiser* that reconstructs clean tokens from noisy ones. We provide details about the noise mechanism in the following paragraph.

Given a tensor $\mathbf{x}_{t,k} = [x_t, ..., x_{t+k-1}]$ and the vocabulary size $V$, we define the discrete diffusion process to each token except the first one using a categorical distribution that has a $\beta(j)$ probability of resampling a category uniformly:

$$q(x^*_{t+j}|x_{t+j}, j) = \mathrm{Cat}(x^*_{t+j}|(1 - \beta(j))x_{t+j} + \beta(j)/V), j \in [2, ..., k-1], \tag{4}$$

where $x^*_j$ is the noisy token and $\mathrm{Cat}$ represents the categorical distribution. Besides, the noise weight $\beta(j)$ is monotonically increased from 0 to 1 within each tensor, i.e., for $j \in [2, ..., k-1]$.

We design a series of scheduling functions $\beta(\cdot)$ as shown in Table 1, to control how the input and noise tokens are fused. These noise scheduling functions include linear, sine, square root, and exponential forms. By modulating the noise intensity across different tokens within a tensor, we simulate a progressive denoising process in autoregressive model training, akin to that in diffusion models. Furthermore, as shown in Figure 3, it is worth noting that we utilize an additional padding token $x_\Delta$, and we ignore the loss calculation at the position of the padding token.

Table 1: Noise scheduling functions.

| Function | Expression |
|---|---|
| *Linear* | $\beta(j) = j/k$ |
| *Sine* | $\beta(j) = sin(\pi j/2k)$ |
| *Square root* | $\beta(j) = \sqrt{j/k}$ |
| *Exponential* | $\beta(j) = j^{\frac{1}{k/2}}$ |

By combining Equation 1 and Equation 2, the overall training objective of TensorAR can be formulated as follows:

$$\mathcal{L}(\theta) = \sum_{i=1}^{T} \sum_{j=1}^{k} \mathbb{E}_{x_{i+j} \sim p_{\text{data}}, x^*_{i+j} \sim q(x^*_{i+j}|x_{i+j}, j)} \left[ w_j \log(p_\theta(x_{i+j}|\mathbf{x}_{<i,k}; c)) \right]. \tag{5}$$

.

Due to the page limit, we provide the pseudo-code of TensorAR during training in the appendix.

### 2.3 RELATION TO OTHER IMAGE GENERATION PARADIGMS

Compared with diffusion models, TensorAR models and trains on image patches in an autoregressive manner, naturally aligning with the discrete sequence modeling paradigm and causal masking used by multimodal large language models. This design enables seamless integration with standard Transformer backbones. Besides, unlike classical diffusion methods that update the entire image at every step, TensorAR updates only the local region covered by the sliding window, preserving iterative refinement while enabling online generation and better scalability. Moreover, unlike standard autoregressive models that generate each patch only once, TensorAR can iteratively refine previously generated patches while producing subsequent content, improving both efficiency and overall visual quality and consistency. *In particular, when $k = 1$, TensorAR reduces to a standard autoregressive model; when $k$ equals the total number of image patches $T$, TensorAR becomes equivalent to a discrete variant of a diffusion process* (with a different generation order, i.e, left-to-right in TensorAR

and random in standard discrete diffusion). During decoding, TensorAR can simultaneously attend to conditions and forthcoming visual information to enforce consistency on earlier content and to complete fine details. Besides, considering the slow inference speed of AR models, especially for large context length, several distillation methods (Liu et al., 2024a; 2025) have been proposed to accelerate the decoding process of AR models with acceptable performance degradation. It will be interesting and promising to integrate these distillation methods and TensorAR to achieve further flexibility in the trade-off between sample quality and sampling speed.

In summary, TensorAR bridges autoregressive and diffusion paradigms, offering a flexible refinement mechanism and a controllable compute–quality trade-off: $k = 1$ provides minimal-latency autoregressive decoding, $k = T$ approximates a discrete diffusion-like multi-step denoising process, and intermediate settings $1 < k < T$ balance efficiency and quality by exploiting future information to iteratively improve previously generated content.

## 3 EXPERIMENTS

### 3.1 EVALUATION ON CLASS-TO-IMAGE GENERATION TASK

We use Fréchet Inception Distance (FID) (Heusel et al., 2017) as our primary metric; we also report Inception Score (IS) (Salimans et al., 2016), Precision and Recall (Kynkäänniemi et al., 2019).

Table 2: Model comparisons on class-conditional ImageNet $256 \times 256$ benchmark. Metrics are Fréchet inception distance (FID), inception score (IS), precision, and recall. "↓" or "↑" indicate lower or higher values are better.

| Type | Model | #Para. | FID↓ | IS↑ | Precision↑ | Recall↑ |
|------|-------|--------|------|-----|-----------|---------|
| Mask AR | MAGVIT-v2 (Yu et al., 2023) | 307M | 1.78 | 319.4 | - | - |
| | MaskBit (Weber et al., 2024) | 305M | 1.52 | 328.6 | - | - |
| | MAR (Li et al., 2024a) | 943M | 1.55 | 303.7 | - | - |
| Casual AR | DART (Gu et al., 2024) | 812M | 3.98 | 256.8 | - | - |
| | RQTran. (Lee et al., 2022) | 3.8B | 3.80 | 323.7 | - | - |
| | ViT-VQGAN-re (Yu et al., 2021) | 1.7B | 3.04 | 227.4 | - | - |
| | SAR-XL (Liu et al., 2024b) | 893M | 2.76 | 273.8 | 0.84 | 0.55 |
| | RandAR-L (Pang et al., 2024) | 1.4B | 2.15 | 322.0 | 0.79 | 0.62 |
| | VAR (Tian et al., 2024) | 2.0B | 1.73 | 350.2 | 0.82 | 0.60 |
| TensorAR | *Open-MAGVIT2* (Luo et al., 2024) | | | | | |
| | Open-MAGVIT2-B ($256 \times 256$) | 343M | 3.08 | 258.3 | 0.85 | 0.51 |
| | +TensorAR | 352M (+2.7%) | 2.91 | 260.2 | 0.86 | 0.50 |
| | Open-MAGVIT2–L ($256 \times 256$) | 804M | 2.51 | 271.7 | 0.84 | 0.54 |
| | +TensorAR | 820M (+2.0%) | 2.35 | 273.4 | 0.84 | 0.53 |
| | *LlamaGEN* (Sun et al., 2024) | | | | | |
| | LlamaGEN-B ($256 \times 256$) | 111M | 5.46 | 193.6 | 0.83 | 0.45 |
| | +TensorAR | 116M (+4.6%) | 4.71 | 225.8 | 0.85 | 0.45 |
| | LlamaGEN-L ($256 \times 256$) | 343M | 3.80 | 248.3 | 0.83 | 0.52 |
| | +TensorAR | 352M (+2.7%) | 2.78 | 254.8 | 0.82 | 0.56 |
| | LlamaGEN-L ($384 \times 384$) | 343M | 3.07 | 256.1 | 0.83 | 0.52 |
| | +TensorAR | 352M (+2.7%) | 2.52 | 258.9 | 0.83 | 0.55 |
| | LlamaGEN-XL ($384 \times 384$) | 775M | 2.62 | 244.1 | 0.80 | 0.57 |
| | +TensorAR | 789M (+1.9%) | 2.29 | 260.4 | 0.81 | 0.59 |
| | LlamaGEN-XXL ($384 \times 384$) | 1411M | 2.34 | 253.9 | 0.81 | 0.60 |
| | +TensorAR | 1432M (+1.5%) | 2.03 | 267.7 | 0.82 | 0.61 |

### 3.1.1 QUANTITATIVE COMPARISON

We evaluate TensorAR on two representative autoregressive (AR) generators—Open-MAGVIT2 (Luo et al., 2024) and LlamaGEN (Sun et al., 2024)—across multiple model scales. Table 2 compares our approach with current state-of-the-art methods. Unless otherwise noted, we set the window size to $k = 4$, use single-layer $Q_{in}$ and $Q_{out}$ modules, and adopt an

exponential scheduling function. TensorAR consistently brings substantial gains over the underlying AR baselines while adding only a small number of parameters. For example, augmenting LlamaGEN-B with TensorAR reduces Fréchet Inception Distance (FID) by 0.71 points. Even on a 1.4B-parameter model, TensorAR achieves a 0.31-point reduction in FID, narrowing the gap to leading diffusion-based models. Moreover, because the auxiliary modules ($Q_{in}$ and $Q_{out}$) are kept fixed across backbones and scales, the relative parameter overhead decreases with model size, i.e., it is approximately inversely proportional to the backbone's overall computational cost.

### 3.1.2 TRAINING FID CURVE

In Figure 5, we plot the training FID curves for TensorAR alongside those from standard fine-tuning of LlamaGEN-B and LlamaGEN-L. Fine-tuning for the same number of steps as used with TensorAR yields no improvement in FID, confirming that TensorAR's gains stem from its design rather than from additional training.

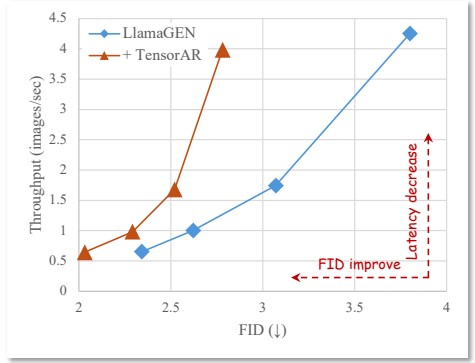

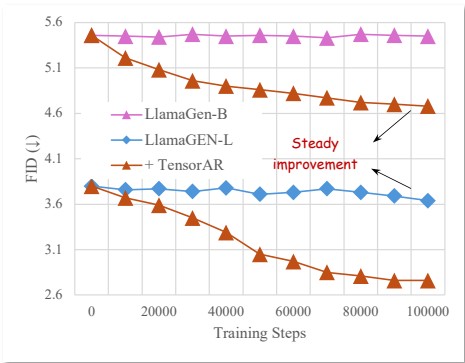

Figure 4: Throughput/FID trade-off. TensorAR consistently improves generation quality with negligible decreases in throughput.

Figure 5: Training FID curves. TensorAR shows steady training dynamics based on two different backbones.

### 3.1.3 THROUGHPUT-FID CURVE

Figure 4 further compares the sampling throughput of TensorAR and LlamaGEN across multiple model sizes. Throughput is measured as the number of samples generated per second (including AR generation and VQ decoding) on a single A100 GPU, using float32 precision and a batch size of 128. Although TensorAR incurs modest additional latency, it delivers substantial FID improvements, yielding a superior efficiency–quality trade-off.

### 3.1.4 IMAGE QUALITY COMPARISON IN THE CLASS-TO-IMAGE GENERATION TASK

We present a qualitative comparison of images generated by LlamaGEN-XXL and TensorAR across four categories. Relative to the base LlamaGEN-XXL, TensorAR produces higher-quality images with richer semantic detail. Additional TensorAR samples are included in the appendix, further demonstrating its ability to generate diverse outputs.

### 3.1.5 VISUAL COMPARISON IN THE TEXT-TO-IMAGE GENERATION TASK

We present a qualitative comparison of images generated by LlamaGEN and TensorAR in the text-to-image generation task. Compared with the base LlamaGEN, TensorAR generates higher-quality images and exhibits more stable instruction-following.

### 3.2 EVALUATION ON TEXT-TO-IMAGE GENERATION TASK

We evaluate TensorAR's text-to-image generation on GenEval (Ghosh et al., 2023) and DPG-Bench (Hu et al., 2024), two benchmarks designed to assess instruction following and compositional

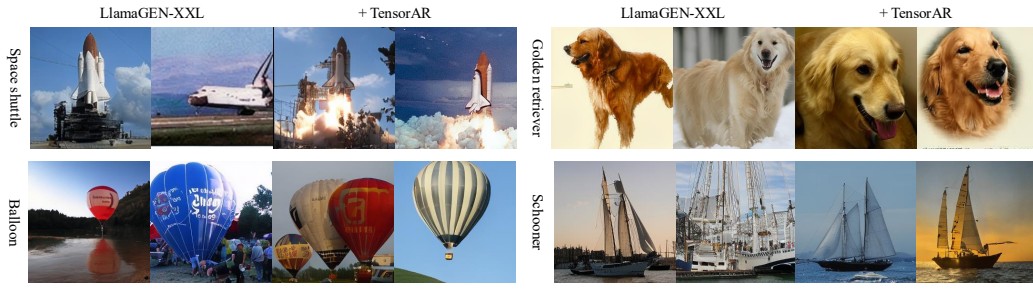

Figure 6: Image generation results comparison. TensorAR can generate high-quality images without loss of diversity. Best viewed in zoom.

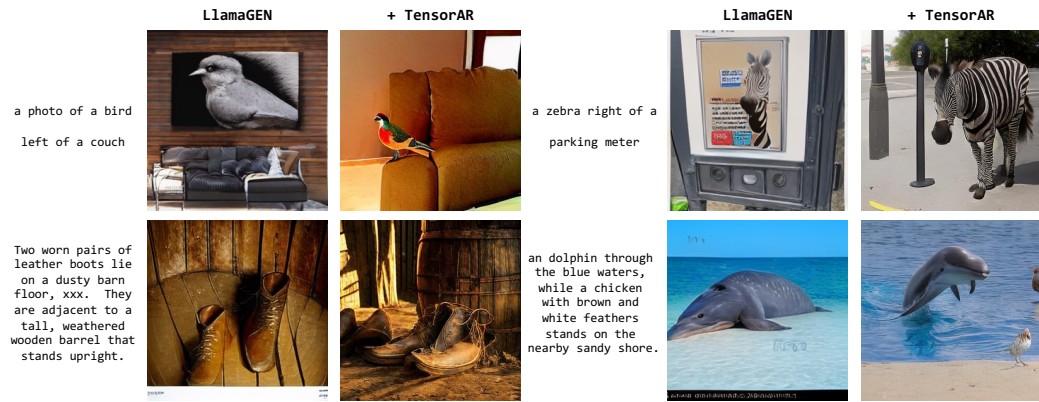

Figure 7: Visual Comparison between LlamaGEN-B and TensorAR in the text-to-image generation task. The two prompts in the first row are selected from the GenEval benchmark, and the other two are selected from the DPG-Bench benchmark. Benefiting from the effectiveness of the proposed TensorAR framework and high-quality data from the BLIP3o dataset, TensorAR can generate more vivid and instruction-following images compared to its baseline counterpart.

alignment. Following the official protocols and metrics, we compare TensorAR with published results for state-of-the-art image generation models, summarized in Table 3 and Table 4. Across both benchmarks, TensorAR delivers consistent gains over its base backbones and remains competitive with state-of-the-art flow-based generators. These findings indicate that integrating TensorAR into existing models enhances instruction-following capability while maintaining strong overall performance. Additional qualitative comparisons of image quality between TensorAR and Janus-Pro-7B are provided in the appendix.

Table 3: Evaluation of text-to-image generation ability on GenEval benchmark. Applying TensorAR brings consistent improvements for different base models.

| Model | Single Obj. | Two Obj. | Counting | Colors | Position | Color Attri. | Overall↑ |
|---|---|---|---|---|---|---|---|
| Emu3-Gen (Wang et al., 2024) | 0.98 | 0.71 | 0.34 | 0.81 | 0.17 | 0.21 | 0.54 |
| DALL-E 3 (Betker et al., 2023) | 0.96 | 0.87 | 0.47 | 0.83 | 0.43 | 0.45 | 0.67 |
| SD3-Medium (Esser et al., 2024) | 0.99 | 0.94 | 0.72 | 0.89 | 0.33 | 0.60 | 0.74 |
| SEED-X (Ge et al., 2024) | 0.97 | 0.58 | 0.26 | 0.80 | 0.19 | 0.14 | 0.49 |
| Show-o (Xie et al., 2024) | 0.95 | 0.52 | 0.49 | 0.82 | 0.11 | 0.28 | 0.53 |
| D-DiT (Li et al., 2025) | 0.97 | 0.80 | 0.54 | 0.76 | 0.32 | 0.50 | 0.65 |
| *TensorAR* | | | | | | | |
| LlamaGen (Sun et al., 2024) | 0.71 | 0.34 | 0.21 | 0.58 | 0.07 | 0.04 | 0.32 |
| + TensorAR | 0.99 | 0.70 | 0.57 | 0.89 | 0.28 | 0.19 | 0.61 |
| Janus-Pro-7B (Chen et al., 2025) | 0.99 | 0.89 | 0.59 | 0.90 | 0.79 | 0.66 | 0.80 |
| + TensorAR | 0.99 | 0.93 | 0.53 | 0.92 | 0.85 | 0.79 | 0.83 |

Table 4: Evaluation of text-to-image generation ability on DPG-Bench benchmark. Applying TensorAR brings consistent improvements for different base models.

| Model | Global | Entity | Attribute | Relation | Other | Overall↑ |
|---|---|---|---|---|---|---|
| Emu3-Gen (Wang et al., 2024) | 85.21 | 86.68 | 86.84 | 90.22 | 83.15 | 80.60 |
| DALL-E 3 (Betker et al., 2023) | 90.97 | 89.61 | 88.39 | 90.58 | 89.83 | 83.50 |
| SD3-Medium (Esser et al., 2024) | 87.90 | 91.01 | 88.83 | 80.70 | 88.68 | 84.08 |
| Hunyuan-DiT (Li et al., 2024b) | 84.59 | 80.59 | 88.01 | 74.36 | 86.41 | 78.87 |
| PixArt-Σ (Chen et al., 2024) | 86.89 | 82.89 | 88.94 | 86.59 | 87.68 | 80.54 |
| *TensorAR* | | | | | | |
| LlamaGen (Sun et al., 2024) | 78.72 | 58.63 | 68.22 | 76.63 | 44.00 | 43.13 |
| + TensorAR | 84.50 | 81.92 | 81.65 | 90.68 | 74.80 | 73.33 |
| Janus-Pro-7B (Chen et al., 2025) | 86.90 | 88.90 | 89.40 | 89.32 | 89.48 | 84.19 |
| + TensorAR | 86.39 | 90.67 | 90.66 | 91.35 | 84.52 | 85.57 |

## 3.3 ABLATION STUDIES

### 3.3.1 DIFFERENT NOISE SCHEDULING FUNCTIONS

As discussed above, the noise scheduling function controls the noise level assigned to each position within a tensor. We evaluate four schedules: linear, sine, square root, and exponential, whose definitions and hyperparameters are summarized in Table 5. We set the base model of all the following ablation studies as LlamaGEN-B in the class-to-image generation task. Across settings, all four schedules yield substantial gains over the base configuration, indicating that TensorAR is robust to the spe-

Table 5: Different noise scheduler functions.

| Model | FID | IS | Precision | Recall |
|---|---|---|---|---|
| Baseline | 5.46 | 193.6 | 0.83 | 0.45 |
| Linear | 4.79 | 218.8 | 0.85 | 0.44 |
| Sine | 4.75 | 221.3 | 0.84 | 0.45 |
| Square root | 4.84 | 214.9 | 0.83 | 0.43 |
| Exponential | 4.71 | 225.8 | 0.85 | 0.45 |

cific choice of schedule. Among them, the exponential schedule achieves the lowest Fréchet Inception Distance (FID), making it a strong default in practice. Overall, these results suggest that the scheduling function is an important factor in TensorAR's performance, with the exponential schedule offering the best efficiency–quality trade-off.

Table 6: Ablation studies on the design of TensorAR.

(a) Different window size $k$

| Model | FID | IS | Precision | Recall |
|---|---|---|---|---|
| Baseline | 5.46 | 193.6 | 0.83 | 0.45 |
| k=2 | 4.78 | 221.3 | 0.84 | 0.45 |
| k=4 | 4.71 | 225.8 | 0.85 | 0.45 |
| k=8 | 4.68 | 226.7 | 0.85 | 0.46 |

(b) Depth of $Q_{in}$ and $Q_{out}$.

| Model | FID | Precision | Recall | Latency |
|---|---|---|---|---|
| Baseline | 5.46 | 0.83 | 0.45 | 0.11 |
| d=1 | 4.71 | 0.85 | 0.45 | 0.12 |
| d=2 | 4.79 | 0.85 | 0.46 | 0.14 |
| d=4 | 4.90 | 0.82 | 0.43 | 0.15 |

### 3.3.2 DIFFERENT WINDOW SIZES

Increasing the window size allows TensorAR to revisit and improve each image token over more steps, which should enhance overall quality. To assess this effect, we vary the window size $k \in \{2, 4, 8\}$ and summarize the results in Table 6a. We observe a monotonic reduction in Fréchet Inception Distance (FID) as $k$ increases, indicating that additional refinement passes are consistently beneficial. Even at $k = 2$—which provides only a single refinement pass per token—TensorAR significantly outperforms the baseline, underscoring the effectiveness of explicit refinement. These findings validate the refinement mechanism as a key contributor to performance. Because larger $k$ entails more sampling steps and thus higher inference cost, practitioners can select $k$ to balance quality and latency, with moderate values offering a favorable trade-off.

### 3.3.3 Depth of $Q_{in}$ and $Q_{out}$

Both $Q_{in}$ and $Q_{out}$ modules are implemented as query transformers, with each layer comprising a cross-attention layer. We investigate the optimal depth for these modules by varying the number of layers $d \in \{1, 2, 4\}$. As reported in Table 6b, $d = 1$ achieves the lowest Fréchet Inception Distance (FID), while increasing to $d = 4$ yields no further improvement. However, considering the quality–latency trade-off, we adopt $d = 1$ as the default, which substantially improves throughput with only a modest impact on image quality. This choice offers a favorable balance for practical deployment.

### 3.4 Visualization of Refinement

As described in Section 2.2, at each decoding step, TensorAR outputs a block of $k$ consecutive tokens. The first token in the block is committed to the final sequence, while the remaining $k - 1$ tokens are provisional and refined in subsequent steps. This commit-and-refine strategy induces a zig-zag, coarse-to-fine progression across positions (Sun et al., 2025): previously emitted tokens (except the first in each block) are iteratively improved as new tokens are introduced. To illustrate this behavior, Figure 8 visualizes the evolution of outputs produced by a Janus-Pro-7B model with a window size of $k = 4$. Applying TensorAR yields higher visual quality and stronger instruction following than the baseline. The images become progressively sharper and semantically richer as refinement proceeds. These qualitative results corroborate the effectiveness of the refinement mechanism. Additional visualizations are provided in the appendix.

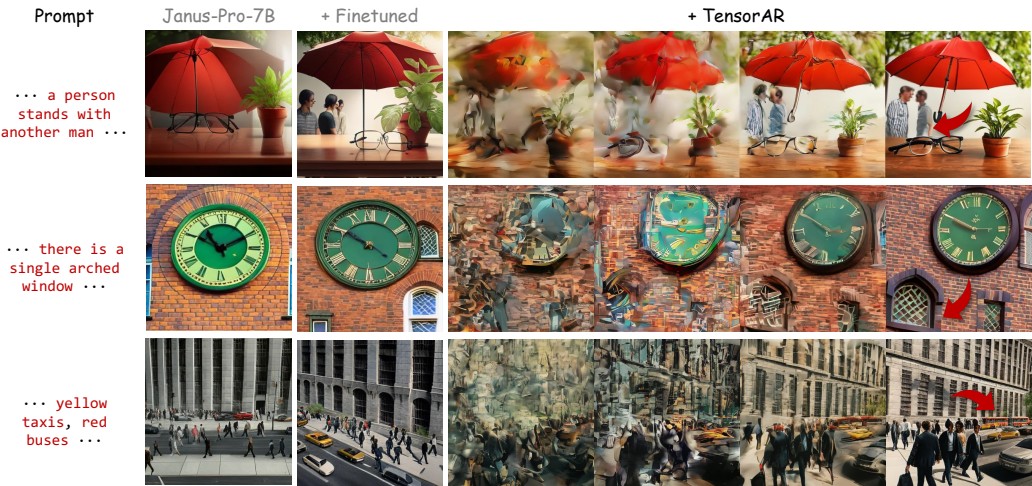

Figure 8: Visualization of the refinement process of TensorAR against its base model: Janus-Pro-7B with a window size $k = 4$. We mark the text that Janus-Pro-7B fails to generate in red and point to the corresponding object generated by TensorAR via a red arrow. All these prompts are from the DPG-Bench benchmark. Best viewed in zoom.

## 4 Conclusion

In this paper, we present TensorAR, to the best of our knowledge, the first visual autoregressive framework that integrates an explicit refinement mechanism into the decoding process. TensorAR extends the conventional next-token prediction paradigm to *next-tensor prediction* by introducing two lightweight plug-in modules, enabling iterative revision of recent outputs. Crucially, it functions as a drop-in augmentation to standard autoregressive transformers, requiring no modifications to the base architecture or changes to the training procedure. Across both class-conditional image synthesis and text-to-image generation, TensorAR delivers consistent improvements in quality, demonstrating the effectiveness of incorporating refinement into visual autoregressive models.

ACKNOWLEDGMENTS

This work was supported by the National Natural Science Foundation of China (No. U24A20291), National Key Research and Development Program of China under Grant (No. 2023YFB4403101), and Sanqin Talent Special Support Plan (No. 2024STZZK09).

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
