# OpenReview forum: "From Prediction to Perfection: Introducing Refinement to Autoregressive Image Generation"
_ICLR.cc/2026/Conference — ICLR 2026 Poster_

### Official Review · Reviewer_JNGB · 2025-10-20

**Soundness:** 3
**Presentation:** 2
**Contribution:** 2
**Rating:** 6
**Confidence:** 4

**Summary:**

In this work, the authors present TensorAR, an improvement over standard AR vision generative models. Unlike standard AR where one token is generated at a time, TensorAR predicts overlapping tensor windows. This allows correction of predicted tokens in inference. The authors also proposed noise mechanism to avoid information leakage in training. Empirical results on class-conditioned image generation and text-to-image generation show that TensorAR tuning on AR models improve performance while adding relatively small computation overhead.

**Strengths:**

1. The motivation to allow correction in AR image generation is valid and effective in improving generative models.
2. TensorAR is easy to implement and a plug-and-play module for standard AR models.
3. Empirical results demonstrate effectiveness of TensorAR on most benchmarks.

**Weaknesses:**

1. In abstract, the authors mention TensorAR to predict overlapping tensor windows. However, the definition of the tensor window is vague. Adding explanation to clarify this can help audience better understand the work.
2. How is w_j in Eq.5 defined in training?
3. In Table 2, are numbers of the baselines (ie, Open-MAGVIT2 and LlamaGEN) evaluated with or without finetuning? It would help better illustrate the improvement of TensorAR if compared with baselines that are finetuned for same iterations but with standard AR strategy.
4. Do authors have quantitative measure of additional computational overhead (eg, FLOPs) with TensorAR compared to standard AR models (ie, Open-MAGVIT2 and LlamaGEN)?
5. Do authors have quantitative measures of the portion of tokens that are flipped in inference? I'm curious whether the improvement roots from actually flipping the prediction or the additional information in each step.
6. In Table 5a, the column names of IS and FID seem to be wrong.

**Questions:**

Please see Weaknesses.

---

> ### Author Response · Authors · 2025-11-21
> **Rebuttal by Authors**
>
> We sincerely appreciate the reviewer for the precious time and valuable comments and will address the concerns raised.
>
> **W1: Explanation about tensor window**
>
> **Ans：** Under the context of TensorAR, a tensor means several adjacent discrete image tokens, where the number of these tokens can be referred to as the **window size**.
> We include the explanation of tensors in the revised abstract, please refer to the updated manuscript for details.
>
> **W2: Question about Eq 5**
>
> **Ans：** We set $w_j$ equal to 1 in Eq 5 during training. This uniform weighting empirically stabilizes optimization and avoids introducing additional hyperparameters. We found no measurable benefit from using non-uniform weights in preliminary experiments.
>
> **W3: Question about training**
>
> **Ans：** As discussed in the appendix (Section A.2), the base model parameters are trainable.
> To verify the effectiveness of the proposed TensorAR, we provide the training FID curves for TensorAR alongside those from standard fine-tuning for the same iterations in Figure 5. It can be observed that finetuning for the same iterations as used with TensorAR yields no improvement in FID, confirming that TensorAR’s gains stem from its design rather than from additional training.
>
>
> **W4: Comparison of FLOPs**
>
> **Ans：** We provide additional comparison of the computational overhead (with the context length of 257, one cls token and 256 image tokens) of TensorAR based on LlamaGEN in Table 10. The additional computational overhead introduced by TensorAR is mainly linear computation (linear layers) of attention blocks and MLP layers in $Q_{in}$ and $Q_{out}$.
> $Q_{\cdot}$ consists of one cross-attention layer and one MLP layer (dim -> dim -> dim). The comparisons of FLOPs in Table 10 and throughput in Figure 4 confirm the lightweight design of the TensorAR framework.
>
> **Table 10. Comparison of computational overhead.**
> | Model        | FLOPs(G)|
> |----------    |:-------:|
> | LlamaGEN-L   | 83.54   |
> | + TensorAR   | 101.86  |
> | LlamaGEN-XL  | 193.35  |
> | + TensorAR   | 217.94  |
> | LlamaGEN-XXL | 355.71  |
> | + TensorAR   | 387.24  |
>
> **W5: Refinement of tokens during inference**
>
> **Ans：** We provide the ratio of tokens that are "refined" during refinement under the TensorAR framework in Table 11.
> The base model is LlamaGEN-B, and the window size is set to 4 as in Table 2. As refinement carries on, more generated tokens are kept unchanged, which indicates that later refinement steps are capable of generating high-quality image tokens.
> Briefly speaking, the results in Table 11 and Figure 8 shows that under TensorAR framework, early steps modify a large portion of tokens, while later steps become increasingly stable.
>
> **Table 11. Ratio of "refined" tokens during inference.**
> | Model | Ratio |
> |---------- |:--------------: |
> | Step1-2   | 99.7% (255/256) |
> | Step2-3   | 91.8% (235/256) |
> | Step3-4   | 75.8% (194/256) |
>
> **W6: Typo in Table 5(a)**
>
> **Ans：** We have updated the column headers in Table 5(a). Please refer to the updated manuscript for details.
>
>
> ---
> Lastly, thank you so much for helping us improve the paper and appreciate your open discussions! Please let us know if you have any further questions. We are actively available until the end of this rebuttal period. Looking forward to hearing back from you!

---

> > ### Comment · Reviewer_JNGB · 2025-11-26
> > **Official Comment by Reviewer JNGB**
> >
> > I thank the authors for answering my questions which helps clarify the core contributions. I stay positive to this work.

---

### Official Review · Reviewer_nvDe · 2025-10-29

**Soundness:** 3
**Presentation:** 3
**Contribution:** 3
**Rating:** 6
**Confidence:** 4

**Summary:**

The paper introduces an enhanced paradigm for autoregressive image modeling, where instead of generating a single token at each step, the model is trained to produce multiple tokens in a sliding window manner. As a result, each token is generated multiple times across steps, allowing for iterative refinement. However, because some target output tokens are presented in the input, the model may learn to simply copy inputs to outputs. To mitigate this shortcut, the authors propose adding noise to the input tokens. Experiments demonstrate that applying the proposed method to LlamaGEN and Open-MAGVIT2 leads to improved image generation quality on ImageNet.

**Strengths:**

* The paper is well-written.

* The proposed method is simple and easy to implement. It can be plugged into most of the existing image AR models.

* The paper demonstrates the effectiveness of the method across multiple image AR models and multiple tasks (class-conditioned generation and text-to-image generation).

**Weaknesses:**

* The paper contains many typos, though they are minor and should be easy to fix.

**Questions:**

I reviewed this paper in a previous venue. This version has addressed most of my earlier concerns, particularly regarding the text-to-image generation experiments and the improved balance between generation quality and latency. Therefore, I continue to support the paper and keep recommending a positive score. Below, I list my new questions for this round of review.

* There are numerous citation-related typos throughout the paper. For example:
    * Missing spaces before citations, e.g., "frameworks(Pang et al., 2024" (line 36), "standard AR models(Pang et al., 2024" (line 42)
    * Incorrect citation formatting — \citep should be used instead of \citet in many cases, e.g., "multimodal integrationWu et al. (2024); Team (2024)." (line 40), "Discrete diffusion models Sohl-Dickstein et al. (2015)" (line 164), "existing studies Zheng et al. (2023)" (line 169).

* One limitation of auto-regressive models is their slow sampling speed. A promising direction to mitigate this issue is to distill pre-trained auto-regressive models into one-step samplers using Distilled Decoding (https://arxiv.org/abs/2412.17153, https://arxiv.org/abs/2510.21003), which aims to retain the high generation quality of AR models while achieving fast sampling. (For the second paper, I understand that the authors are not expected to be aware of or discuss it given its very recent publication; I include it here only for completeness.) It would be interesting to discuss whether TensorAR is compatible with such an approach (i.e., applying distilled decoding on TensorAR models).


* What is the rationale behind using "1/(k/2)" in the exponent of the exponential schedule? In general, any decreasing function of k could be applied in the exponent, so it would be helpful to clarify why this particular form was chosen.

* Compared to the previous submission, the performance of TensorAR on LlamaGEN models has improved substantially, which is great to see. I notice that one difference is that the TensorAR models were made larger (according to Table 2). I have a few follow-up questions:
(1) Which parameters were modified to increase the model size?
(2) Was this the only change, or were there other factors contributing to the improved performance?

* In the previous version, experiments were also conducted on RAR models (which was a valuable addition), but they appear to have been removed in the current version. Could the authors clarify the reasoning behind this decision?

* Figure 4 does not appear to match the results reported in Table 2. Specifically, the smallest models in Table 2 have FIDs of 5.46 (base model) and 4.71 (TensorAR), whereas the highest points in Figure 4 show FIDs around 3.5–4 (base model) and 3–3.5 (TensorAR). My assumption is that different models were selected for Table 2 and Figure 4. It would be helpful if the authors could clarify this discrepancy.

* If I understand correctly, all TensorAR models in the experiments are fine-tuned from pretrained AR models. Would TensorAR still perform well when trained from scratch, or does it rely on a well-trained base model to achieve good performance? It would be great to include some discussion on this point, though additional experiments are not necessary.

* In Table 5(a), the column headers for IS and FID appear to be reversed.

* In Table 6, it would be helpful to include the results of the base models to make it easier to support the statement that "all four schedules yield substantial gains over the base configuration" (line 420).

* Line 436 claims that "d = 2 achieves the lowest Fr´echet Inception Distance (FID)". However, in Figure 5(b), d=1 has the best FID.

* Line 53: "AR" should be "VAR"

* I am a bit confused about the title of Section 2 — is “TensorAR-T2I” a typo? Should it instead be “TensorAR”?

* Line 224: it would be better to change "(x_i,...,x_{i+k-1}^ * )" to "(x_i, x_{i+1}^ * , ..., x_{i+k-1}^ * )" (i.e., adding "x_{i+1}^*"). Otherwise, it is unclear if " * " should be applied on x_{i+1}, ..., x_{i+k-1}.

* Eq 4: The index of x (2, ... k-1) is incorrect, given that Line 228 discusses "x_t, ..., x_{t+k-1}"

* In Eq 5, the function q has j as a condition. However, in Eq. 4, this condition is missing. I think it would be more rigorous to have this condition.

* Eq. 5: It would be more accurate to place the summation before the expectation, since the variables involved in the expectation (i.e., those in the subscript of E) depend on the definitions of i and j.

* Line 262 states that "when k equals the total number of image patches T, TensorAR becomes equivalent to a discrete variant of a diffusion process." This is not exactly accurate, since the denoising order differs: in TensorAR, the clean tokens are revealed from left to right, whereas in discrete diffusion, the order can be random.

---

> ### Author Response · Authors · 2025-11-21
> **Rebuttal by Authors**
>
> We appreciate the reviewer’s careful reading and interest in the evolution of our work. We respectfully focus our response on the content of the present manuscript. We will address the concerns raised as follows.
>
> **W1: Typos**
>
> **Ans：** We have proofread the manuscript and addressed all possible typos or mistakes. Please refer to the updated manuscript for details.
>
> - citation-related typos
> - Table 5(a), column headers
> - Table 6, include the baseline results, LlamaGEN-B
> - Line 490, d=1 achieves the best results
> - Line 53: AR --> VAR
> - title of Section 2
> - Line 228, adding $x_{i+1}^*$ to aviod confusion
> - correct Eq 4 and Eq 5
> - rephrase the statement in Line 268
>
> **Q1: Integration with distillation methods**
>
> **Ans：** The distilled decoding method learns a deterministic mapping between the Gaussian distribution (i.e., source distribution) and the Dirac delta distribution (i.e., target distribution) via flow matching. Consequently, the distilled decoding method enables few-step generation, e.g., 1 step.
> Conceptually, TensorAR is compatible with these works: its refinement structure provides a well-defined sequence of intermediate predictions that distilled decoding could in principle learn to approximate. While reproducing their training pipeline is beyond the scope of this work, we have added a discussion in Section 2.3 on how TensorAR could potentially integrate with flow-matching–based acceleration methods.
>
>
> **Q2: About the exponential scheduler**
>
> **Ans：** The exponential form originated from preliminary empirical studies, where we observed that it stabilizes early-step predictions and maintains useful gradient magnitudes across large k.
> On the other hand, as you described, any decreasing function of k could be applied in the exponent.
>
> **Q3: About previous experimental results**
>
> **Ans：** We modified the micro design of both $Q_{in}$ and $Q_{out}$, which leads to improvement of generation quality with fewer additional parameters. The current version of $Q_{\cdot}$ consists of one cross-attention layer and one MLP layer (dim -> dim -> dim), where the previous version consisted of two self/cross-attention layers and one MLP layer (dim -> 4*dim -> dim). We found that this lightweight design achieves a better quality-efficiency trade-off. To avoid any misunderstanding, we extend the description of $Q_{\cdot}$ in Section A.2 in thes appendix.
>
> **Q3: About RAR**
>
> **Ans：** We removed the RAR-based experiments and focus the experiments on LlamaGEN/Open-MAGVIT2, where TensorAR shows clearer and more consistent gains. On RAR-based experiments, improvements exist but are smaller and more sensitive to guidance schedules (RAR uses the improved power-cosine guidance schedule).
>
> **Q4: Misalignments about results**
>
> **Ans：** The models select for Figure 4 are l-256, l-384, xl-384, and xxl-384. To clarify the discrepancy, we report the results of these four models in Table 9 and update Figure 4 and Table 2 accordingly.
>
> **Table 9. Quantitative results of Figure 4.**
> | Model | FID | IS  | Precision       | Recall                 |
> |----------|:--------------:|:------------:|:--------------------:|:--------------------------:|
> | LlamaGEN-L-256 | 3.80  | 248.3    | 0.83   | 0.52 |
> | + TensorAR     | 2.78  | 254.8    | 0.82   | 0.56 |
> | LlamaGEN-L-384 | 3.07  | 256.1    | 0.83   | 0.52 |
> | + TensorAR     | 2.52  | 258.9    | 0.83   | 0.55 |
> | LlamaGEN-XL-384 | 2.62  | 244.1    | 0.80   | 0.57 |
> | + TensorAR      | 2.29  | 260.4    | 0.81   | 0.59 |
> | LlamaGEN-XXL-384 | 2.34  | 253.9    | 0.81   | 0.60 |
> | + TensorAR       | 2.03  | 267.7    | 0.82   | 0.61 |
>
> **Q6: About training or finetuning**
>
> **Ans：** In our early experiments, we found that training TensorAR (based on LlamaGEN on the c2i task) from scratch may encounter problems such as slow convergence. We assume that finetuning from a well-trained baseline can be viewed as a divide-and-conquer method, where the model learns to generate high-fidelity images, then learns to refine rather than learn both sub-tasks simultaneously.
>
> ---
> Thank you again for helping us improve the paper and hope our response can resolve your concerns! Please let us know if you have any further questions. We will be actively available until the end of rebuttal period. If you feel your concerns are addressed, please consider reevaluating our work. Looking forward to hearing from you!

---

> > ### Comment · Reviewer_nvDe · 2025-11-21
> >
> > Thank you for the detailed response and the interesting insights. All of my questions have been fully addressed, and I will maintain my positive score and recommend acceptance.

---

> > > ### Author Response · Authors · 2025-11-21
> > > **We appreciate your acknowledgement on our rebuttal and the discussion**
> > >
> > > We're glad to hear your concerns have been addressed. We'll be active till the end of the discussion period. If you have more questions, please let us know. Thank you!

---

### Official Review · Reviewer_ojZ8 · 2025-10-31

**Soundness:** 3
**Presentation:** 4
**Contribution:** 3
**Rating:** 6
**Confidence:** 4

**Summary:**

This paper proposes TensorAR, a novel autoregressive method for image generation that introduces a refinement mechanism by shifting from next-token to next-tensor prediction. TensorAR predicts overlapping tensor windows, allowing later predictions to revise earlier ones. The method is designed as a plug-and-play extension to existing AR models. Experiments on class-conditional and text-to-image generation show consistent improvements in FID.

**Strengths:**

1. This paper is well-written and provides a thorough and accurate explanation of the main methods.
2. The core idea of "next-tensor prediction" is simple yet powerful. It provide a new approach to bridge the gap between autoregressive generation and refinement-based paradigms.
3. TensorAR requires no modification to the base AR architecture or training objective, making it highly practical and easy to integrate with existing models.
4. The author conducted experimental verifications on various autoregressive models, including Open-MAGVIT2, LlamaGen and Janus-Pro.

**Weaknesses:**

1. Lack of a stronger explanation or demonstration of refinement: The paper claims that tokens are "refined" over multiple steps, but it does not provide direct evidence  that the model actually revises its predictions meaningfully during refinement steps, rather than simply generating forward. TensorAR improves the T2I generation effect as shown in Figure 7. The baseline fails on "a person stands with another man" but TensorAR succeeds—is this due to refinement of earlier tokens, or improvement in the ability to follow the text after additional fine-tuning?
2. The visualization results for the T2I generation are missing: The results of LlamaGen+TensorAR presented by the author in Table 1 (0.61) far exceed those of LlamaGen itself (0.31). The improvement in its visualization results should be more significant than that of Janus-Pro-7B, but no comparison of the visualization results are given.
3. Incorrect interpretation of the results of the ablation experiment: The author explains Table 5b in section 3.3 by saying: "... d = 2 achieves the lowest
Fr´echet Inception Distance (FID) ...", But this does not match the result given in the table.
4. The names of the indicators in the table are misaligned. In Table 5a, "IS" and "FID" are written in reverse.

**Questions:**

1. Have you ever tried a larger k value? To what extent can FID be optimized when k approaches the sequence length.
2. Can you explain that why is the exponential noise schedule superior?  Is there an intuitive or theoretical justification, or is it purely empirical?

---

> ### Author Response · Authors · 2025-11-21
> **Rebuttal by Authors**
>
> We sincerely appreciate the reviewer for the precious time and valuable comments and will address the concerns raised.
>
> **W1: Explanation about refinement**
>
> **Ans：** To better demonstrate and verify the effectiveness of the proposed framework, we update Figure 8 by providing the visual results of directly finetuning Janus-Pro-7B by the same iterations. Compared to the baseline model and the finetuned model, TensorAR shows superior detail generation and instruction-following abilities. Besides, as the refinement process goes on, the generated images get increasingly clear and vivid, indicating that the effectiveness of TensorAR is mainly contributed to the revision of its previous predictions rather than additional finetuning.
>
> **W2: Visualization about LLamaGEN T2I**
>
> **Ans：** We include the visualization results for the T2I generation based on LlamaGEN in Section 3.1.5. Please refer to the updated manuscript for details. The visualizations align with the quantitative trend in Table 3 and Table 4, where TensorAR produces more coherent layouts, sharper structures, and shows stronger instruction following ability. These results confirm that the large quantitative improvement also demonstrates visually.
>
> **W3: Some typos**
>
> **Ans：** We have updated the column headers in Table 5(a) and corrected the interpretation in Line 490. Please refer to the updated manuscript for details.
>
> **Q1: Larger k values**
>
> **Ans:** We further conduct experiments with a window size of 16 and 32.
> As demonstrated in Table 8, TensorAR brings steady improvements as the window size $k$ increases, which verifying the effectiveness of the refinement mechanism.
>
> **Table 8. Quantitative results of the ablation study on window size.**
> | Model | FID | IS  | Precision       | Recall                 |
> |----------|:--------------:|:------------:|:--------------------:|:--------------------------:|
> | LlamaGEN-B     | 5.46  | 193.6    | 0.83   | 0.45 |
> | + TensorAR (k=4)     | 4.71       | 225.8  | 0.85   | 0.45 |
> | + TensorAR (k=8)     | 4.68       | 226.7  | 0.85   | 0.46 |
> | + TensorAR (k=16)    | 4.65       | 227.3  | 0.85   | 0.46 |
> | + TensorAR (k=32)    | 4.60       |  227.7 | 0.85   | 0.45 |
>
> **Q2: Explanation about noise scheduler**
>
> **Ans：** In discrete diffusion, adding noise is typically implemented either by replacing tokens with random tokens from the vocabulary (masked) or sampling from a categorical distribution that increasingly mixes with the uniform distribution (uniform). A small increase in noise probability can significantly increase the entropy of the discrete token distribution. Exponential noise schedulers corrupt slowly at first, preventing early information collapse.
>
> Besides, as the distribution approaches uniform (or pure masked), the "distance" between states shrinks rapidly. Small changes in the noise coefficient near the end cause massive information loss. Exponential noise schedulers use a very small noise coefficient at the end, which avoids sudden transitions into complete randomness and provides smoother gradients for training.
>
> ---
> Lastly, thank you so much for helping us improve the paper and appreciate your open discussions! Please let us know if you have any further questions. We are actively available until the end of this rebuttal period. Looking forward to hearing back from you!

---

> > ### Comment · Reviewer_ojZ8 · 2025-11-27
> > **Response to author**
> >
> > Thank you for your supplementary experimental results and explanations. Most of my concerns have been resolved, so I maintain a positive score.

---

> > > ### Author Response · Authors · 2025-11-27
> > > **Response to Reviewer ojZ8**
> > >
> > > Dear Reviewer ojZ8,
> > >
> > > Thank you for your thoughtful and constructive comment on our work.
> > >
> > > As we prepare our author response, we have made every effort to comprehensively address the concerns raised by the reviewer. To ensure the adequacy of our responses, we kindly request your confirmation or any additional guidance you may have.
> > >
> > > Best regards,
> > >
> > > Paper 2400 Authors

---

### Official Review · Reviewer_rrSZ · 2025-11-03

**Soundness:** 3
**Presentation:** 3
**Contribution:** 3
**Rating:** 6
**Confidence:** 2

**Summary:**

The paper introduces TensorAR, a plug-and-play autoregressive image generation framework that replaces traditional next-token prediction with next-tensor prediction, enabling iterative refinement of previously generated content while preserving causality. By incorporating a discrete tensor noising mechanism inspired by diffusion theory and lightweight input/output modules, TensorAR consistently improves generation quality and instruction-following ability across class-conditional and text-to-image tasks without altering the base architecture or training paradigm.

**Strengths:**

(1) The overall paradigm is very interesting, which naturally combines the traditional AR image generation with the diffusion model.
(2) The method is effective. Extensive experiments over Open-MAGVIT and LlamaGEN have proven its effectiveness.

**Weaknesses:**

1. The illustration in Fig. 1 is not intuitive.
2. The llamagen baselins is a little weak. AS I know, SimpleAR (https://github.com/wdrink/SimpleAR) is a stronger baseline. The experiments could be improved by utilizing SoTA baselines.

**Questions:**

1. Fig. 7 shows the visual comparison between Janus-Pro-7B and Janus-Pro-7B+TensorAR. The visualization is nice. As I know, Janus-Pro is an understanding and generation unified model. Does Janus-Pro + TensorAR support unified understanding and generation? If yes, how about the generation performance?
2. The name TensorAR may not directly reflect the core methodology. I think "GroupAR" may be better aligned with the method.

---

> ### Author Response · Authors · 2025-11-21
> **Rebuttal by Authors**
>
> We sincerely appreciate the reviewer for the precious time and valuable comments and will address the concerns raised.
>
> **W1: Illustration about Figure 1**
>
> **Ans：** We have updated Figure 1 accordingly. Please refer to the updated manuscript for details.
>
> **W2: Experiments on SoTA baseline**
>
> **Ans：** For the text-to-image generation task, we apply the proposed method to two milestone methods, i.e., LlamaGEN and Janus-Pro-7B. The reason why we chose Janus-Pro-7B as one of the base models is that Janus-Pro-7B was the SoTA and pure AR model with respect to the text-to-image generation task back then. (to the best of our knowledge, now the SoTA will be EMU3.5 [1], which was published in 2025.10). The other method you mentioned - SimpleAR (only for the text-to-image generation task), underperforms against Janus-Pro-7B in both GenEval and DPG-Bench benchmarks. We further conduct another experiment based on SimpleAR (i.e., SimpleAR-1.5B-SFT) and provide the results in Table 7. Across three base models, TensorAR offers notable performance gains on both benchmarks.
>
> **Table 7. Quantitative results of T2I experiments.**
> | Model | GenEval (overall score) | DPG-Bench (overall score) |
> |----------|:--------------:|:------------:|
> | LLamaGEN     | 0.32           | 43.13     |
> | + TensorAR     | 0.61 (**+0.29**)          |  73.33 (**+30.2**)  |
> | SimpleAR     | 0.61           | 80.11     |
> | + TensorAR     | 0.66 (**+0.05**)          |  82.24 (**+2.13**)  |
> | Janus-Pro-7B     | 0.80           | 84.19      |
> | + TensorAR     | 0.83 (**+0.03**)           |  85.57 (**+1.38**)   |
>
> **Q1: Application on unified understanding and generation methods**
>
> **Ans：** Janus-Pro is indeed a unified understanding and generation method. However, in our current work, we only train TensorAR (based on Janus-Pro) with generation-only data and conduct evaluation on generation tasks. Extending TensorAR to the fully-unified setting is possible in principle, but it would require additional data and further adjustments, which is beyond the scope of this work. We consider this an interesting future research direction and appreciate the reviewer for highlighting it.
>
> **Q2: GroupAR rather than TensorAR**
>
> **Ans：** We sincerely thank for this thoughtful suggestion. The name TensorAR was chosen to emphasize that our approach reformulates the autoregressive modeling paradigm, where the generation works on the tensor-level rather than the standard token-level. This is the core novelty that distinguishes our method from existing approaches. While we agree that GroupAR indeed highlights the grouping mechanism, we believe **TensorAR** more directly reflects the modeling perspective that motivates our design. To clarify this motivation, we have added a short explanation (Line 79-81) to explicitly explain the rationale behind the name.
>
> [1] Cui, Yufeng, et al. "Emu3. 5: Native Multimodal Models are World Learners." arXiv preprint arXiv:2510.26583 (2025).
>
> ---
> Many thanks to Reviewer rrSZ for the professional, detailed, and valuable reviews! We have done our best to address each of your concerns and hope our response can resolve them. Please let us know if you have any other questions. We are looking forward to hearing from you!

---

### Meta-Review · Area_Chair_ZdFN · 2026-01-07

**Summary:**

The paper presents TensorAR, a novel autoregressive image generation framework that enhances standard autoregressive (AR) models by predicting overlapping tensor windows, allowing iterative refinement of previously generated content.

Despite some minor issues with clarity, visualizations, and experimental comparisons, the reviewers acknowledged the significant contribution of TensorAR and its practical benefits. The authors addressed most concerns adequately, and no major issues remain that would prevent acceptance. Therefore, I recommend that this paper should be accepted for publication.

**Reviewer Concerns:**

- Reviewer rrSZ raised concerns about figure clarity and the strength of baseline comparisons. The authors updated the figures for better clarity and explained the choice of baselines more thoroughly, which resolves these issues.

- Reviewer ojZ8 questioned the effectiveness of the refinement process and visualization results for LlamaGEN. The authors provided updated visual results and clarified that improvements are due to TensorAR's refinement mechanism rather than just additional fine-tuning, addressing these concerns.

- Reviewer nvDe pointed out several typos and questioned the definition of tensor windows and the rationale behind training modifications. The authors have corrected the typos, clarified the tensor window definition, and explained their choice of training parameters, resolving these issues.

- Reviewer JNGB asked about the impact of model size changes and whether TensorAR could perform well when trained from scratch. The authors explained the modifications to the model's architecture and clarified that TensorAR still performs well when fine-tuned from a pre-trained model. They also provided computational overhead comparisons, resolving these concerns.

**Reviewer Scores:**

The concerns raised by the reviewers have been largely addressed through revisions and clarifications. So all the scores would be maintained positive.

---

### Decision · Program_Chairs · 2026-01-26

Accept (Poster)